# Correlates of the Mental Fitness of Female High School Freshmen: Focus on Multidimensional Empathy and Brain Function

**DOI:** 10.3390/ijerph17218290

**Published:** 2020-11-09

**Authors:** Hee Wook Weon, Jiyoung Lim, Hae Kyoung Son

**Affiliations:** 1Department of Brain and Cognitive Science, Seoul University of Buddhism, Seoul 08559, Korea; soojiwon@hanmail.net; 2College of Nursing, CHA University, Pochen-shi 11160, Korea; limjy62@cha.ac.kr; 3Department of Nursing, Eulji University, Seongnam-shi 13135, Korea

**Keywords:** adolescent, brainwaves, empathy, students, women’s health

## Abstract

We examined the association between multidimensional empathy, brain function, and mental fitness and identified correlates of mental fitness. In total, 146 female high school freshmen from a South Korean school participated in this cross-sectional study. Data were collected from March to April 2019, using a self-report questionnaire and quantitative electro-encephalographic data (QEEG). Instruments included the Interpersonal Reactivity Index and the Mental Fitness Scale, to access multidimensional empathy and mental fitness. Prefrontal cortex brain function was assessed with the brain quotient measure from the QEEG during free time after school. Data were analyzed using descriptive statistics, Pearson’s correlation coefficient, and multiple regression analysis. Mental fitness had statistically significant relationships with multidimensional empathy (r = 0.36, *p* < 0.001) and brain quotient (r = 0.23, *p* = 0.005). Demographic factors affecting mental fitness included satisfaction with school life (β = 0.23, *p* = 0.001) and economic status (β = 0.17, *p* = 0.024). Factors from the subscales of multidimensional empathy included perspective taking (β = 0.26, *p* = 0.001), fantasy (β = 0.22, *p* = 0.004), and personal distress (β = −0.19, *p* = 0.010); and the brain function factor was brain quotient (β = 0.14, *p* = 0.038). The explanatory power of the model was 49.4% (*F* = 14.44, *p* < 0.001). There is a need for a concrete and objective understanding of mental fitness in adolescents to develop intervention programs for freshmen with various maladaptation problems.

## 1. Introduction

Adolescence is a transient period between childhood and adulthood during which young people experience significant physical and emotional transformations and prepare themselves for their social roles as adults [1]. With growing demand for social developmental tasks, adolescents are increasingly responsive to changes and stimuli in their surroundings. In particular, high school freshmen are subject to multiple burdens, such as rapid physical growth, exploring career paths, and adapting to a new environment in high school. In this tough period of environmental adaptation, adolescents are biologically vulnerable to unstable or stressful events that affect their mental fitness. Therefore, society strives to develop alternative programs to improve adolescents’ mental fitness [2]. However, Korean high school education still focuses on educational aspects, such as preparation for the college/university entrance examination and career decision-making. This makes it difficult for adolescents to attend to mental fitness within an overly competitive environment.

In recent years, aggressive and antisocial issues such as bullying and violence among youths have emerged, which might be associated with adolescents’ failure to maintain mental fitness [3,4,5]. According to previous research, adolescent girls suffer more serious effects from internalized problems than adolescent boys, such as withdrawal, depression, social immaturity, thinking problems, and concentration deficits [6]. Since adolescent girls who are antisocial tend to be more sensitive to disruptions in social environments, particularly dysfunctional family backgrounds, they are at a higher risk of suffering from various problems as adults, hence negatively affecting the next generation [6]. Consequently, there is a compelling need to provide adolescent girls with adequate intervention programs to promote mental fitness [7,8,9].

Mental fitness is defined as the changeable capacity to utilize resources and skills to psychologically adapt to environmental challenges or advantages, to meet psychological needs [10]. The mental fitness concept emerged as an analogy to physical fitness in the early 1960s and was conceived by drawing parallels between physical and mental fitness—positive mental fitness can be attained through systematic mental training; just as physical fitness can be attained through systematic physical training [11]. A new approach proposed by the Positive Psychology movement breaks away from the conventional disease model and seeks to adopt a more active understanding of mental fitness in people who are not mentally ill [2]. Unlike conventional psychological and psychiatric concepts, the conceptualization of mental fitness was premised upon the idea that positive emotions and negative emotions are independent of each other; reducing negative emotions does not automatically lead to corresponding increases in positive emotions, but might increase neutral emotions [2,10]. 

Mental fitness is based on a higher dimensional understanding, such as empathy with others or reflecting on one’s personal life. In relation to mental fitness, lack of empathy can lead to difficulty expressing emotions or impede understanding of our own or others’ minds. In general, empathy is the ability to recognize others’ emotional state and its meaning, and to use that knowledge to communicate with others, which enables individuals to infer their own and others’ state of mind, including beliefs, desires, and intentions [12]. Accordingly, empathy strengthens social cohesion as a factor that precedes the development of social cognitive skills or social intelligence, and is an essential factor in successful social interactions [13,14].

Empathy is typically conceptualized as a multidimensional construct [8], consisting of perspective taking, fantasy, empathic concern, and personal distress. These are considered key components in practical and theoretical perspectives [15]. According to Davis [15], perspective taking refers to a tendency to spontaneously adopt the psychological point of view of others; and fantasy is defined as the tendency to transpose oneself imaginatively into the feelings or actions of fictitious characters in books, movies, and plays, as if they were real. Empathic concern refers to other-oriented feelings of sympathy and concern for people in distress, whereas personal distress involves self-oriented feelings of personal anxiety and unease in tense interpersonal relationships.

This study simultaneously investigated the prefrontal cortex brain function of adolescents, by employing quantitative electro-encephalogram (QEEG). This study differs from previous research, in that, it applied a more objective approach that overcomes the limitations of psychometric measurement. Furthermore, it sought to understand brain functions through QEEG, which contribute to objectively understanding subjective perceptions and emotions closely associated with mental fitness and multidimensional empathy, by analyzing the differences in brainwave electrical signals (i.e., differences in the α, β, δ, and θ waves emitted from the left and right hemispheres of the brain, when the brain cells communicate with each other) [16]. As the brain’s executive center is responsible for advanced thinking skills, the prefrontal cortex forms personality, regulates the depth of emotions (such as anger and aggression) and controls logical thinking. QEEG has a long history and its clinical usefulness has gradually become well-known [17,18,19,20]. The rapid development of desktop computers in the 1990s placed QEEG technology in the hands of clinicians. In addition, the powerful desktop computers of today have paved the way for new and faster methods of analysis for brain activity [20]. QEEG defines the mathematical processing of digitally recorded EEG, in order to highlight specific waveform components, transform the EEG into a format or domain that elucidates relevant information, or associate numerical results with the EEG data for subsequent review or comparison [21]. Accordingly, this study aimed to analyze multidimensional empathy, brain function, and mental fitness in female high school freshmen and to determine the correlates of their mental fitness based on multidimensional empathy and brain function. 

### Aims

This study’s concrete objectives were to:Examine multidimensional empathy, brain function, and mental fitness in female high school freshmenExamine the correlation between multidimensional empathy, brain function, and mental fitness in female high school freshmenExamine the effects of multidimensional empathy and brain function on mental fitness in female high school freshmen

## 2. Materials and Methods

### 2.1. Study Design

A cross-sectional study design was used to examine correlates of mental fitness in female high school freshmen, based on multidimensional empathy and brain function.

### 2.2. Setting and Sample

Study participants were 17 years old female freshmen attending J high school in S city in South Korea. Among female freshmen who voluntarily agreed to participate after understanding the study objectives and procedures, those who could read and complete the questionnaire were enrolled as participants. Exclusion criteria included literacy problems, as because data were collected through a self-reported questionnaire survey, and those under psychiatric care (drugs or therapeutic interventions), which might affect multidimensional empathy brain function, and mental fitness. However, none of the participants met any of these exclusion criteria.

The study sample size was calculated using G* Power software (G*Power 3.1.7, Heinrich-Heine-University, Düsseldorf, Germany). The minimum sample size was calculated to be 138 at a target significance level of 0.05, power of 0.95, median effect size of 0.15, and five predictors. However, the total number of participants was set at 153, assuming a 10% predicted dropout rate for the QEEG brain function analysis. After excluding seven participants’ data, because they did not participate in the QEEG data collection, the data from 146 participants were used in the data analysis.

### 2.3. Ethical Considerations

Prior to initiating recruitment and data collection, the study was approved by the Institutional Review Board of S University (No.27004121AN01-201903-HR-055-02). Additionally, the investigator obtained prior permission from the participating school regarding the conduct of the study. As the participants were minors, written consent was obtained from their parents or legal guardians. The investigator used a structured information sheet explaining the research objectives, data collection procedures, benefits of participation in the study, and the right to freely withdraw from the study at any time, so that participants could fully consider whether to participate. After obtaining informed consent from each person willing to participate, the researcher provided each with a copy of the informed consent document for record-keeping. The questionnaire was collected by a trained researcher immediately after completion, separate from the informed consent form, to ensure anonymity and personal information protection. All participants were given the QEEG results and a gift (pen), as a token of appreciation.

### 2.4. Instruments

#### 2.4.1. Demographic Characteristics

Demographic characteristics included participants’ religion, birth order, health status, academic achievement, satisfaction with school life, economic status, residential type, and blood type.

#### 2.4.2. Mental Fitness

The level of mental fitness was measured with the Mental Fitness Scale developed by a Korean college of medicine to measure the mental fitness of the general public [2]. The 22-item instrument consisted of five subfactors—mental energy (6 items), empathetic communication (6 items), flexibility (5 items), self-assurance (2 items), and self-understanding (3 items). The items were rated on a 5-point Likert scale (0 = ‘not at all’; 4 = ‘very much so’), with a higher score indicating a higher level of mental fitness. The Cronbach’s α of the Mental Fitness Scale was 0.91 for the overall scale and 0.86, 0.79, 0.71, 0.75, and 0.67 for each subfactor [2]. In this study, Cronbach’s α was 0.91 for the overall scale, and 0.83, 0.68, 0.66, 0.66, and 0.68 for each subfactor. The investigator received prior permission from the author to use the instrument.

#### 2.4.3. Multidimensional Empathy

Multidimensional empathy was measured with the Korean version of the Interpersonal Reactivity Index (IRI) developed by Davis [15]. The IRI is a widely-used individual difference measure of empathy based on a multidimensional construct. This 28-item scale consists of four independently applicable subscales—perspective taking, fantasy, empathic concern, and personal distress. Each subscale consisted of seven items, which were rated on a 5-point scale (0 = ‘This sentence does not describe me well’; 4 = ‘This sentence describes me very well’), whereby a negative sentence is scored in reverse fashion. Accordingly, the IRI provides scores on four different components of empathy. Higher total scores indicate higher levels of multidimensional empathy by subscale. In development of the K-IRI, Cronbach’s α was 0.80 for the overall scale, and 0.61, 0.81, 0.73, and 0.71 for the subscales of perspective taking, fantasy, empathic concern, and personal distress, respectively. In this study, Cronbach’s α was 0.80 for the overall scale, and 0.66, 0.80, 0.76, and 0.70 for the subscales, respectively. The investigator received prior permission from the author to use the instrument.

#### 2.4.4. Brain Function

Brain function was measured using QEEG by the investigator, who is a board-certified neurofeedback therapist. QEEG is a noninvasive, easy, comfortable, and inexpensive way to check brain status [18]. For QEEG, a 2-Channel Neurofeedback System (NeuroHarmony M, Braintech Corp. Korea) measured the bilateral prefrontal cortex with active electrodes placed on Fp1 and Fp2 and at the left earlobe, with a reference electrode in the user-friendly format instrument of a headband. After connecting a 2-channel EEG measurement system to a PC, using a dry electrode attached to the headband, EEG analysis was performed in the order of first-round open eye 40 seconds, closed eye 40 seconds, and second-round open eye 40 seconds on the prefrontal lobes, followed by measuring the Self-Regulation Quotient for relaxation, attention, and concentration, for one minute each. The brain function indices analyzed by QEEG in this study are presented in Table 1.

The Basic Rhythm Quotient (BRQ), which indicates the levels of brain development, aging, and stability, represents brainwaves (ranging from ≤ 40 μV to ≥ 100 μV) that are dominant in a resting state with eyes closed. If the left and right BRQs have α wave frequency, the brain activity at the corresponding age is optimal in the range of 9–11 Hz. The Self-Regulation Quotient (SRQ), which measures the ability to control the autonomic nervous system that regulates brain relaxation, attention, and concentration, is the most basic measure of brain health and activity, ranging from ≤ 40 μV to ≥ 100 μV. When the frequencies were related to relaxation, attention, and concentration (i.e., α waves), the sensorimotor rhythm, and the low β waves were all well-balanced (≥ 30 μV each), and the brain was considered to be healthy and active. The ATtention Quotient (ATQ), which is the level of arousal of the left and right hemispheres of the brain, ranged from ≤ 20 μV (extremely distracted state) to ≥ 80 μV (fully aroused state). The ACtivation Quotient (ACQ), which represented the level of brain activity, identified mental activity, thinking ability, and behavioral tendency, ranging from ≤ 20 μV (functional deficiency) to ≥ 80 μV (maximum activity). The Emotion Quotient (EQ), which indicated the balance of emotional stability and instability, ranged from ≤ 20 μV (very unstable state) to ≥ 80 μV (very stable state). The Stress Resistance Quotient (SQ), which indicates physical and mental fatigue related to internal and external environmental factors, ranges from ≤ 20 μV (pathological state) to ≥ 80 μV (optimal health). The Correlation Quotient (CQ), which indicated the balance between the left and right hemispheres of the brain, ranged ≤ 40 μV (completely asymmetrical) to ≥ 90 μV (perfectly symmetrical). The Brain Quotient (BQ), which evaluated the overall brain functioning based on these seven EEG indices, ranged from ≤ 40 μV (lowest brain functioning) to ≥ 90 μV (highest brain functioning) [22,23]. The reliability of this instrument was established at 0.916 (*p* < 0.001), relative to the Grass System (USA), as an EEG measurement device. 

### 2.5. Data Collection

Data collection was carried out from March to April 2019, at the beginning of the semester. In this period, the characteristics of freshmen who were trying to adjust to high school life, were most conspicuous. After obtaining prior permission from the school, the investigator visited the school outside of school hours to administer the questionnaire and measure EEG. A trained researcher distributed the questionnaire and collected it ten minutes later, separately from the consent form.

### 2.6. Data Analysis

Data analysis was performed using IBM SPSS Statistics ver. 22.0 (IBM Co., Armonk, NY, USA). Demographic characteristics were analyzed using descriptive statistics, reliability was assessed using Cronbach’s alpha coefficient, intervariable correlations were calculated using Pearson’s correlation coefficient, and intervariable effects were calculated using multiple regression. The correlation between measurement variables was determined by the magnitude of the correlation coefficient, and the effect between variables was considered to have statistical significance at a *p* value < 0.05. For accurate analysis of QEEG, a spectral power analysis of intensity, frequency, and ratio in brain waves was performed using Fast Fourier Transform (FFT) of the Neurofeedback System (Braintech Corp., Korea).

## 3. Results

### 3.1. Participants

Participants included 146 female high school freshmen. They were all aged 17 years. Analysis of their demographic characteristics (Table 2) showed that Christianity was the most frequent religion with 31.5% (*n* = 46). In terms of birth order, 17 (11.6%) were an only child, 37 (25.3%) first child, 79 (54.1%) second child, and 13 (8.9%) were a third child or later. “Good” was the most frequent answer for perceived health status (*n* = 55; 37.7%), followed by “fair” (*n* = 47; 32.3%), and “very good” (*n* = 38; 26.0%). The most frequent response regarding academic achievement in the last semester was “good” (*n* = 54; 37.0%), followed by fair (*n* = 48; 32.9%). In satisfaction with school life, the most frequent response was “satisfied” (*n* = 74; 50.7%), followed by “very satisfied” (*n* = 36, 24.7%) and “neutral” (*n* = 30, 20.5%). “Neutral” was the most frequent response for economic status (*n* = 72; 49.3%). All participants (*n* = 146, 100%) responded that they lived at home with family. The most frequent blood type was A (*n* = 50; 34.2%), followed by B (*n* = 43; 29.5%), O (*n* = 42; 28.8%), and AB (*n* = 11; 7.5%). Of the demographic characteristics, factors affecting mental fitness were found to be health status (*F* = 9.95, *p* < 0.001), academic achievement (*F* = 2.60, *p* = 0.039), satisfaction with school life (*F* = 7.12, *p* < 0.001), and economic status (*F* = 8.21, *p* < 0.001).

### 3.2. Variables

The mean values of variables are shown in Table 3. The mean values for the mental fitness and multidimensional empathy were 58.23 ± 12.78 and 74.33 ± 11.78 points, respectively. The QEEG results revealed that brain development and stability were “fair” with BRQ (Left/Right) measures 69.62 ± 13.24 μV and 70.21 ± 11.07 μV, respectively. The left/right *α*-wave frequencies—the BRQ-related frequencies—were 9.72 ± 0.95 Hz and 9.71 ± 0.95 Hz, respectively, which indicated an adequate age-specific level of brain activity, with 17 participants (11.6%) showing 8 Hz. The mean for SRQ was 56.33 ± 19.23 μV, indicating that self-regulation ability to maintain states of relaxation, attention, and concentration was low. A detailed SRQ-related frequency analysis showed the mean for relaxation, attention, and concentration to be 25.82 ± 6.40 μV, 18.18 ± 7.38 μV, and 19.09 ± 7.35 μV, respectively. This was lower than their respective ideal standard mean values (27 μV, 24 μV, and 25 μV, respectively). The ATQ (Left/Right) measures (35.20 ± 13.49 μV and 34.45 ± 13.47 μV) indicated a distracted state of brain arousal. With ACQ (Left/Right) of 36.34 ± 10.89 μV and 36.73 ± 11.40 μV, brain activity was observed to be “poor.” EQ, with mean 74.94 ± 4.87 μV, showed an emotionally stable state. The SQ (Left/Right), with 69.66 ± 14.36 μV and 68.35 ± 14.87 μV, indicated “good” in terms of physical and mental fatigue. With mean 78.07 ± 14.01 μV, the CQ indicated a state of balanced development of the left and right brain hemispheres, showing a balance between the amplitude and phase of the brain. The mean BQ, which evaluated the overall brain function based on the above-mentioned seven indices, was 56.96 ± 7.36 μV, which fell in the semi-poor category. 

### 3.3. Relationships between Variables

Correlations between multidimensional empathy, brain function, and mental fitness are shown in Table 4. Mental fitness was significantly positively correlated with multidimensional empathy (r = 0.36, *p* < 0.001), for the subscales—perspective taking, fantasy, empathic concern, and personal distress (*p* < 0.001). Mental fitness was significantly positively correlated with BQ, an overall evaluation of brain function (r = 0.23, *p* = 0.005); specifically, with the left BRQ (r = 0.20, *p* = 0.015), right BRQ (r = 0.19, *p* = 0.022), right ATQ (r = 0.19, *p* = 0.020), EQ (r = 0.23, *p* = 0.005), and left SQ (r = 0.17, *p* = 0.040).

### 3.4. Correlates of Mental Fitness

Table 5 shows the results of testing the basic assumptions through regression analysis of demographic characteristics (health status, academic achievement, satisfaction with school life, and economic status), the subscales of multidimensional empathy (perspective taking, fantasy, empathic concern, and personal distress), and the overall measure of brain function (BQ) to identify the correlates of mental fitness. The Durbin–Watson statistic was 1.84, showing no autocorrelation, tolerance exceeded 0.01 in all variables, and the variance inflation factor (VIF) values ranged between 1.09 and 1.68, demonstrating that there were no multicollinearity problems between the chosen explanatory variables. Thus, the final regression model was statistically significant (F = 14.44, *p* < 0.001), with significant demographic characteristics including satisfaction with school life (β = 0.23, *p* = 0.001) and economic status (β = 0.17, *p* = 0.024); significant multidimensional factors including empathy, perspective taking (β = 0.26, *p* = 0.001), fantasy (β = 0.22, *p* = 0.004), and personal distress (β = −0.19, *p* = 0.010); and BQ (β = 0.14, *p* = 0.038), all verified as correlates of mental fitness, with an explanatory power of 49.4%.

## 4. Discussion

The strength of this study was the investigation of correlates of mental fitness from multiple perspectives, with various QEEG-based brain function indices as physiological indicators, and the multidimensional factor of empathy measured with self-report questionnaires. The brain’s electrical signals contained regular patterns that might be better understood by their spectral content. Bursts of sinusoidal waves occurred and reoccurred in a predictable fashion, and these bursts corresponded with mental states [20]. Especially, QEEG reports typically include displays of mean spectral magnitude or power for multiple frequency bands. This information might be provided as means and percentile change from typical and atypical EEG data, are used to support one’s conclusions and training recommendations, culminating in a report [20].

First, the multidimensional empathy subscales of “perspective taking,” “fantasy,” and “personal distress” were identified as correlates of the mental fitness of female high school freshmen. However, “empathic concern,” which involves other-oriented feelings of sympathy and concern, did not have an impact on mental fitness. According to Davis [15], the relationships between the empathic concern scale measuring a specific type of emotional response and other measures of emotionality will depend on the precise nature of these emotionality measures. This finding is supported by the results of research conducted by Shanafelt et al. [24] on residents in physician training, where the level of empathy was compared between residents with or without high mental well-being. This study found that higher mental fitness was associated with significantly higher scores in perspective taking as the cognitive empathic domain, while no significant differences were observed in empathic concern as the emotive empathic domain. Medical residents’ situation—working long shifts at large academic hospitals to master the knowledge of their specialty while rapidly adapting to patient characteristics whenever they rotate—is similar to the situation of high school freshmen, who are busy adapting to their new school life and spending the majority of their time at school, while studying to gain knowledge. Given the critique by many researchers that empathy-related studies overemphasize the cognitive aspects of empathy, such as taking others’ perspectives, and overlooking the ability to recognize and understand emotions experienced by individuals [15], it is necessary to conduct replication studies that explore empathic concern in various stressful life stages, such as that of female high school freshmen.

In particular, the mean values for the multidimensional empathy subscales were 17.94 ± 4.29 points for perspective taking, 19.45 ± 5.22 points for fantasy, 19.30 ± 4.70 points for empathic concern, and 17.59 ± 4.48 points for personal distress. Tracy et al. [14] reported that an analysis of empathy among female students in high school and university, yielded the following multidimensional empathy subscale means—empathic concern 19.52 points and personal distress 14.28 points. Compared with these results, the present study found higher scores for personal distress subscales. A previous study [14] found significant cultural differences on empathy, with an East Asian group showing lower scores than a Western group. In the result of Tracy et al. [14], relative to East Asian adolescents, Western adolescents reported greater levels of empathic concern and less personal distress when confronted with another person’s negative emotional state. The higher levels of empathic concern and lower levels of personal distress in Western adolescents, suggest a more other-oriented emotional response to another’s distress. Further examination should explore detailed characteristics of empathy in adolescents and examine whether the situational effect of female high school freshmen is reflected in the results. On this note, a follow-up study with male high school freshmen is proposed, drawing on the results of earlier studies [14,25,26] indicating that females showed higher levels of empathy than their male counterparts in a comparison of multidimensional empathy.

Our study verified positive correlations with mental fitness demonstrated by the multidimensional empathy subscales of “perspective taking,” “fantasy,” and “empathic concern.” On the other hand, “personal distress” was observed to be negatively correlated with mental fitness. Empathy, based on the understanding and acceptance of others’ feelings and views plays a crucial role in gaining mental fitness by fostering successful social interactions and prosocial characteristics, such as social cohesion, and reducing or suppressing negative aspects, such as conflict and aggression [14]. Personal distress is negatively related to social functioning; persons prone to anxiety and discomfort in emotional social settings will have more difficulty establishing and maintaining rewarding social relationships than persons not prone to such feelings [15]. Since empathy naturally develops through continuous interactions with others during adolescents’ developmental process, there is a need to understand the effects of multidimensional empathy on mental fitness.

In this study, the QEEG results in female high school freshmen showed the mean BRQ (left/right) to be 69.62 ± 13.24 μV and 70.21 ± 11.07 μV, respectively, suggesting a “fair” state of brain development and stability. The left/right BRQ frequency, which is a critical index of brain development levels during adolescent brain function changes, was measured at 9.72 ± 0.95 Hz and 9.71 ± 0.95 Hz, respectively. Brainwave frequencies increase with brain weight increases through age-based brain development, and the degree of brainwave alteration begins to decrease with the onset of adulthood. High school students in late adolescence attained approximately the same brainwave frequencies as adults. The reference frequency for the brain activity rate of the corresponding age (i.e., 9 to 11 Hz for α waves in the 15 to 45 age range), was appropriate, and the participants were generally found to have the age-specific frequencies. However, in the detailed analysis, 17 participants (11.6%) had 8 Hz in the left/right BRQ frequency, suggesting slow brain development or activity, given that it is a frequency appropriate for children up to 10 years or elderly persons from 60 onwards [27]. Since brain development and stability affect the brain activity rate, that is, how fast the brain works, it is proposed that cases with low BRQ values be screened and examined through a detailed analysis of brainwave sub-variables, with the aim of developing intervention programs.

Participants’ mean SRQ was 56.33 ± 19.23 μV, suggesting a low level of self-regulation. The SRQ, which is the first measure examined when analyzing the brain function, is a basic measure of brain health and activity that identifies whether three states of brain activity—relaxation, attention, and concentration—are adequately self-regulated in the waking state [23]. The mean values for relaxation, attention, and concentration were 25.82 ± 6.40, 18.18 ± 7.38, and 19.09 ± 7.35 μV, respectively, which were lower than their respective standard means of 27, 24, and 25 μV. In general, a low relaxation index was associated with exhaustion or insomnia; a low attention index indicates distraction, lack of sociality, and asocial status; and a low concentration index indicated weak concentration, lack of confidence, and a lack of momentum. The closer the values for relaxation, attention, and concentration were to the standard mean values, the better. In this regard, participants’ mean values for the attention and concentration indices were somewhat lower than that of the relaxation index. This might be associated with the period-specific situation in which high school freshmen need to respond sensitively to the changes and stimuli of their surroundings and strive to adapt to them. This suggests that individuals need to acquire basic developmental skills, such as concentration, activeness, and momentum as well as sociality and sociability, necessary for the transition from childhood to adulthood.

The ATQ (left/right) means were 35.20 ± 13.49 μV and 34.45 ± 13.47 μV, respectively, which indicated a distracted state of brain arousal. A low ATQ means strong θ waves, which is associated with attention deficiency and memory loss, which lower learning ability [28]. An extremely low ATQ is generally associated with Attention Deficit Disorder, Attention Deficit Hyperactivity Disorder, or tick disorders [29]. Participants’ ATQ values suggest that they tend to be distracted or their learning ability is impaired by memory loss in a state of rather high physical tension and fatigue.

ACQ (left/right), the measure for brain activity, had corresponding mean values of 36.34 ± 10.89 μV and 36.73 ± 11.40 μV, respectively, which fall into the category of “poor” brain activity. What is noteworthy is that the participants’ left ACQ and right ACQ were at similar levels. In general, when the left brain is activated, rational, logical, mathematical, and verbal skills are developed. In contrast, when the right brain is activated, emotional, intuitive, comprehensive, and artistic skills are developed [30]. This is consistent with the participants’ situation, given that they attended a general high school, not a specialized (arts/sports/foreign language/technology) high school, and data collection was conducted at the onset of the first high school semester, after they learned a wide variety of subjects in middle school. In replication studies, various brain activity indices from a variety of participants should be compared, to identify factors that influence left and right brain activities, to enable generalization of the results.

The EQ measure, with a mean of 74.94 ± 4.87 μV, indicated an emotionally stable state. EQ signifies the character of the brain, identifying the degree of bias in the emotion tendency (i.e., bright-and-brisk or dark-and-depressed mindsets). Participants’ EQ values suggest that they were emotionally stable without being biased towards manic or depressive episodes. The mean values of SQ (left/right), 69.60 ± 14.36 μV and 68.35 ± 14.87 μV, respectively, indicated “good” health status with high resistance against physical and mental stress. In general, physical stress displays physical tension and mental stress displays psychological tension, anxiety, and excitement. The higher the physical and mental stress, the higher the fatigue, and the lower the resistance to illness; therefore, higher SQ means stronger resistance. These SQ-related results are coherent in that participants’ level of satisfaction with school life was generally high and satisfaction with school life affects mental fitness. The participants demonstrated an emotionally stable state and good resistance against physical and mental stress.

The CQ mean was 78.07 ± 14.01 μV, indicating a state of balanced development in the left and right brain hemispheres. An excessive imbalance between the left and right brain hemispheres triggers various problems of imbalance including physical imbalance, as well as linguistic, emotional, and activity disorders [27]. Participants’ general state of balanced left and right brain hemisphere development with no significant imbalance suggests that they were equipped to start and continue attending general education in high school. That is, the sample characteristics were well-reflected, which highlights the importance of assessing differences in sample characteristics to facilitate future research that is generalizable.

Mean BQ, which evaluates overall brain function based on the seven indices, was 56.96 ± 7.36 μV, falling in the semi-poor category and thus identified as a correlate of mental fitness. In comparison to a study conducted by Byun and Park [23], in which the mean BQ values for male and female high school students were 58.63 ± 2.65 μV and 60.65 ± 9.09 μV, respectively, mean BQ in the current study’s participants was lower, suggesting that participants’ brain function was slightly affected during the period of adaptation to a new school environment. This predicted reduced brain activity for learning ability during this time. BQ provides exponential numerical information similar to IQ or EQ, and it is necessary to determine the difference in brain function resulting from characteristics, such as gender and academic achievement, in future research, especially during a specific period, such as the freshmen year. The mean values for mental fitness in this study (58.23 ± 12.78 points) was higher than that in healthy adults (50.5 ± 10.4 points) or psychiatric patients (37.6 ± 12.2 points) at the time of development of the Mental Fitness Scale. This demonstrates that it is necessary to adopt a more active understanding when defining mental fitness in mentally fit people. To identify an individual’s level of mental fitness based on this study’s findings, an understanding of individual empathy and objective analysis of brain function is needed. When developing intervention programs to promote mental fitness, content designed to enhance multidimensional empathy and brain function needs to be included.

## 5. Limitations

This study had several limitations. Participants’ self-assessment of multidimensional empathy and mental fitness using structured questionnaires, poses limitations such as subjectivity. However, the IRI is one of the most widely used self-report measures of empathy in circulation and has both good internal and external validity [15]. Additionally, this study used brain function indices along with a self-report measure. Considering the problem of the short-term measurement, this study finally suggested repeated measurement of brain function indices among high school students, with a variety of characteristics.

## 6. Impact Statement

This study suggested that high schools incorporate mental fitness interventions into the learning process so that high school freshmen can strengthen their multidimensional empathy and brain function in adolescence.

## 7. Conclusions

This study analyzed the correlates of mental fitness with a focus on multidimensional empathy and brain function. It is significant for examining the correlates of mental fitness in female high school freshmen, through a neuroscientific approach, utilizing various quantitative electro-encephalogram (QEEG)-based brain function indices along with a self-report measurement. The authors suggest that intervention programs be established based on objective indicators for treating various maladaptive behaviors in adolescents, including high school freshmen. We also propose exploring brain function with multi-channel brain signals for EEG measurement, and extending the method used in this study using a 2-channel system.

## Figures and Tables

**Table 1 ijerph-17-08290-t001:** Distribution of brain function.

Quotients	Ranges						
	Very Poor	Poor		Fair	Good	Very Good	
BRQ (L/R)	≤40	40–60		60–80	80–100	≥100	
SRQ	≤40	40–60		60–80	80–100	≥100	
ATQ (L/R)	≤20	20–40		40–60	60–80	≥80	
ACQ (L/R)	≤20	20–40		40–60	60–80	≥80	
EQ	≤20	20–40		40–60	60–80	≥80	
SQ (L/R)	≤20	20–40		40–60	60–80	≥80	
CQ	≤40	40–55		55–70	70–90	≥90	
**Interpretation**	**Very poor**	**Poor**	**Semi-poor**	**Fair**	**Good**	**Semi-good**	**Very good**
BQ	≤40	40–50	50–60	60–70	70–80	80–90	≥90

Note: BRQ = Basic Rhythm Quotient, SRQ = Self-Regulation Quotient, ATQ = ATtention Quotient, ACQ = ACtivation Quotient, EQ = Emotion Quotient, SQ = Stress resistance Quotient, CQ = Correlation Quotient, BQ = Brain Quotient, L = Left, and R = Right.

**Table 2 ijerph-17-08290-t002:** Participant demographic characteristics (*n* = 146)

Characteristics	Categories	Frequency (%)	F (*p*)
Religion	Christianity	46 (31.5)	0.51 (0.673)
	Buddhism	3 (2.1)	
	Catholicism	9 (6.2)	
	None	8 (60.3)	
Birth order	Only child	17 (11.6)	1.42 (0.241)
	1 st	37 (25.3)	
	2 nd	79 (54.1)	
	3 rd and over	13 (8.9)	
Health status	Very good	38 (26.0)	9.95 (<0.001)
	Good	55 (37.7)	
	Neutral	47 (32.2)	
	Poor	6 (4.1)	
	Very poor	0 (0.0)	
Academic achievement	Very good	15 (10.3)	2.60 (0.039)
	Good	54 (37.0)	
	Neutral	48 (32.9)	
	Poor	20 (13.7)	
	Very poor	9 (6.2)	
Satisfaction of school life	Very satisfied	36 (24.7)	7.12 (<0.001)
	Satisfied	74 (50.7)	
	Neutral	30 (20.5)	
	Unsatisfied	5 (3.4)	
	Very unsatisfied	1 (0.7)	
Economic status	Very good	10 (6.8)	8.21 (<0.001)
	Good	51 (34.9)	
	Neutral	72 (49.3)	
	Poor	10 (6.8)	
	Very poor	3 (2.1)	
Residential types	Home with family	146 (100.0)	-
ABO type	A type	50 (34.2)	0.20 (0.894)
	B type	43 (29.5)	
	AB type	11 (7.5)	
	O type	42 (28.8)	

**Table 3 ijerph-17-08290-t003:** Level of variables (*n* = 146)

Measures	Mean ± *SD*	Range/Meaning
Mental fitness	58.23 ± 12.78	0–88
Mental energy	15.86 ± 4.78	0–24
Empathetic communication	17.01 ± 3.75	0–24
Flexibility	13.99 ± 3.49	0–20
Self-assurance	3.29 ± 2.04	0–8
Self-understanding	8.09 ± 2.63	0–12
Multidimensional empathy	74.33 ± 11.78	0–112
Perspective-taking	17.94 ± 4.29	0–28
Fantasy	19.45 ± 5.22	0–28
Empathic concern	19.30 ± 4.70	0–28
Personal distress	17.59 ± 4.48	0–28
Brain function		
BRQ (L/R)	69.62 ± 13.24/70.21 ± 11.07	Fair
SRQ	56.33 ± 19.23	Poor
ATQ (L/R)	35.20 ± 13.49/34.45 ± 13.47	Poor
ACQ (L/R)	36.34 ± 10.89/36.73 ± 11.40	Poor
EQ	74.94 ± 4.87	Good
SQ (L/R)	69.60 ± 14.36/68.35 ± 14.87	Good
CQ	78.07 ± 14.01	Good
BQ	56.96 ± 7.36	Semi-poor

Note: BRQ = Basic Rhythm Quotient, SRQ = Self-Regulation Quotient, ATQ = Attention Quotient, ACQ = Activation Quotient, EQ = Emotion Quotient, SQ = Stress resistance Quotient, CQ = Correlation Quotient, BQ = Brain Quotient, L = Left, and R = Right.

**Table 4 ijerph-17-08290-t004:** Correlations between measures (*n* = 146)

Measures	Mental Fitness
	r (*p*)
Multidimensional empathy	0.36 (<0.001) **
Perspective taking	0.52 (<0.001) **
Fantasy	0.43 (<0.001) **
Empathic concern	0.39 (<0.001) **
Personal distress	−0.34 (<0.001) **
Brain function	
BRQ (L/R)	0.20 (0.015) */0.19 (0.022) *
SRQ	0.06 (0.463)
ATQ (L/R)	0.16 (0.055)/0.19 (0.020) *
ACQ (L/R)	0.05 (0.544)/0.05 (0.554)
EQ	0.23 (0.005) **
SQ (L/R)	0.17 (0.040) */0.13 (0.116)
CQ	0.00 (0.969)
BQ	0.23 (0.005) **

Note: * *p* < *0*.05, ** *p* < *0*.01, BRQ = Basic Rhythm Quotient, SRQ = Self-Regulation Quotient, ATQ = ATtention Quotient, ACQ = ACtivation Quotient, EQ = Emotion Quotient, SQ = Stress resistance Quotient, CQ = Correlation Quotient, BQ = Brain Quotient, L = Left, and R = Right.

**Table 5 ijerph-17-08290-t005:** Correlates of mental fitness (*n* = 146)

Factors	B	S.E.	β	t	*p*	Tolerance	VIF
(Constant)	45.51	9.42		4.83	<0.001		
Health status	1.65	1.10	0.11	1.51	0.135	0.75	1.33
Academic achievement	−0.03	0.92	0.01	0.04	0.971	0.85	1.17
Satisfaction of school life	3.77	1.15	0.23	3.28	0.001	0.81	1.24
Economic status	2.46	1.08	0.17	2.29	0.024	0.77	1.30
Multidimensional empathy							
Perspective taking	0.74	0.22	0.26	3.36	0.001	0.69	1.45
Fantasy	0.50	0.17	0.22	2.90	0.004	0.71	1.40
Empathic concern	0.21	0.21	0.08	1.00	0.318	0.60	1.68
Personal distress	−0.50	0.19	−0.19	−2.62	0.010	0.79	1.27
Brain function							
BQ ^1^	0.22	0.11	0.14	2.10	0.038	0.92	1.09

Adjusted R^2^ = 0.494, F = 14.44 (*p* < 0.001), Durbin-Watson = 1.84. ^1^ BQ = Brain Quotient

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
