# Peer review of "Correlates of the Mental Fitness of Female High School Freshmen: Focus on Multidimensional Empathy and Brain Function"

_ijerph, 2020, doi:10.3390/ijerph17218290_

Round 1

Reviewer 1 Report

The purpose of this study was to analyze the correlations between multidimensional empathy, brain function, and mental fitness in female high school freshmen, and to determine the factors affecting their mental fitness based on multidimensional empathy and brain function. The findings make contributions to the research in mental fitness. However, I have a few comments and suggestions before it is published.

This is a cross-sectional study. So suggest not to use the terms like “factor” or “predict” which imply a causal relationship. Suggest to use “correlates” or “be associated with”. “Potential factor” is also acceptable if you provide a strong justification for the potential causal relationship.

The authors sometimes used mental fitness but sometimes used mental health. I am wondering whether they are the same and interchangeable?

Is there any reason for why the authors used subscales of empathy but total score of the brain function in the regression analyses?

The authors mentioned that “Compared with these results, the present study found higher scores for personal distress subscales. Previous study [14] found significant cultural differences on empathy, with an East Asian group showing lower scores than a Western group.” Any explanation for the potential cultural difference? Please further discuss.

Please provide a paragraph to discuss the limitations of this study.

It is plausible that mental fitness can affect brain function. The authors need to discuss this possibility. In addition, the authors are suggested to provide more robust justifications for the effects of brain function on mental fitness.

Author Response

Thank you for your valuable feedback for the improvement of this article.

We have revised the manuscript.

Point 1: This is a cross-sectional study. So suggest not to use the terms like “factor” or “predict” which imply a causal relationship. Suggest to use “correlates” or “be associated with”. “Potential factor” is also acceptable if you provide a strong justification for the potential causal relationship.

Response 1: Thank you for your valuable input. Upon general review of the manuscript, I used the terms “correlates” or “be associated with”. Thank you.

Point 2: The authors sometimes used mental fitness but sometimes used mental health. I am wondering whether they are the same and interchangeable?

Response 2: Upon general review of the manuscript, current study was revised the term “mental health” to the term “mental fitness”. Thank you again for such detailed feedback.

Point 3: Is there any reason for why the authors used subscales of empathy but total score of the brain function in the regression analyses?

Response 3: According to Davis, empathy is typically conceptualized as a multidimensional construct. Accordingly, the instrument to measure empathy is a widely used individual difference measure of empathy based on a multidimensional construct. Moreover, BQ evaluates overall brain function based on the seven indices. In this study, brain function means BQ as a representative index. A detailed description of the multidimensional empathy has been included on page 2 & page 4. Thank you.

Point 4: The authors mentioned that “Compared with these results, the present study found higher scores for personal distress subscales. Previous study [14] found significant cultural differences on empathy, with an East Asian group showing lower scores than a Western group.” Any explanation for the potential cultural difference? Please further discuss.

Response 4: Thank you for this helpful suggestion. I have added an additional sentence depicting the research finding on page 10: “In the result of Tracy et al. [14], relative to East Asian adolescents, Western adolescents reported greater levels of empathic concern and less personal distress when confronted with another person’s negative emotional state. The higher levels of empathic concern and lower levels of personal distress in the Western adolescents suggest a more other-oriented emotional response to another’s distress”. Thank you.

Point 5: Please provide a paragraph to discuss the limitations of this study.

Response 5: Thank you for this valuable suggestion. The section ‘5. Limitations’ has been added on page 12. Thank you again for such detailed feedback.

Point 6: It is plausible that mental fitness can affect brain function. The authors need to discuss this possibility. In addition, the authors are suggested to provide more robust justifications for the effects of brain function on mental fitness.

Response 6: Upon general review of the manuscript, current study was added to the reference list and the contents were revised accordingly. After careful consideration of the feedback, further explanations of the ages of participants has been added to the “4. Discussion” section to enhance readers’ understanding. Thank you.

Sincerely.

Reviewer 2 Report

Intro - I have an issue with the term "mental fitness". The authors report on why this is used (ala physical fitness) and provide references, though these appear to be "old" from 2013 and 2015.  I wonder whether the term "resilience" is preferable.  I also refer to this below in 2.4.2

The background would benefit from a discussion about why quantitative EEG is useful contribution to the study.  How is QEEG different to regular EEG?  The authors quote that "QEEG has a long history of"...this is vague and needs elaboration.  

For international readers, it would be useful to put the ages of what "Freshmen" refers to.

2.4.2 It's likely cultural-related but I have a  problem with the term "mental fitness".  It appears that in Korea, "mental fitness" is a "thing" but I feel that the nomenclature is wrong or inappropriate.  This is clearly not the case in Korea as they have developed their own "Mental fitness" scale. 

2.5 Data collection "are most conspicuous" - should be in Methods section.

Discussion - The authors provide a standard last sentence (lines 412 - 415) about why this study is important.  Further detail would enhance this.

Author Response

Thank you for your valuable feedback for the improvement of this article.

We have revised the manuscript.

Point 1: Intro - I have an issue with the term "mental fitness". The authors report on why this is used (ala physical fitness) and provide references, though these appear to be "old" from 2013 and 2015.  I wonder whether the term "resilience" is preferable.  I also refer to this below in 2.4.2.

Response 1: Thank you for your valuable input. Fundamentally, resilience refers to ability to maintain or regain mental fitness, despite experiencing adversity. The definition of resilience expanded to become protective and vulnerability forces at multiple levels of influence – culture, community, family and the individual. Accordingly, although they look like, the term “mental fitness” in this study are different from the term “resilience”. After careful consideration of the feedback, further explanations of the term “mental fitness” have been added to the main text to enhance readers’ understanding with the latest. If anything else is required with regard to this, I will be glad to consider the same. Thank you for your valuable feedback for the improvement of this article.

Point 2: The background would benefit from a discussion about why quantitative EEG is useful contribution to the study.  How is QEEG different to regular EEG?  The authors quote that "QEEG has a long history of"...this is vague and needs elaboration.  

Response 2: After careful consideration of the feedback, further explanations of the ages of participants has been added to the “1 Introduction” section to enhance readers’ understanding. Thank you.

Point 3: For international readers, it would be useful to put the ages of what "Freshmen" refers to.

Response 3: After careful consideration of the feedback, further explanations of the ages of participants has been added to the “2.2 Setting and Sample” section and “3.1 Participants” section to enhance readers’ understanding. Thank you.

Point 4: 2.4.2 It's likely cultural-related but I have a  problem with the term "mental fitness".  It appears that in Korea, "mental fitness" is a "thing" but I feel that the nomenclature is wrong or inappropriate.  This is clearly not the case in Korea as they have developed their own "Mental fitness" scale. 

Response 4: I agree with you. I have revised the sentence: The level of mental fitness was measured with the Mental Fitness Scale developed by a Korean college of medicine to measure the mental fitness of the general public [2]. Thank you.

Point 5: 2.5 Data collection "are most conspicuous" - should be in Methods section.

Response 5: I agree with you. I have revised the section “2.5 Data collection”. Instead, I have added the following sentences to the ‘2.4.4 Brain function’ section on page 4: “After connecting a 2-channel EEG measurement system to a PC, using a dry electrode attached to the headband, EEG analysis was performed in the order of first-round open eye 40 seconds, closed eye 40 seconds, and second-round open eye 40 seconds on the prefrontal lobes, followed by measuring the Self-Regulation Quotient for relaxation, attention, and concentration, for one minute each”.  Thank you.

Point 6: Discussion - The authors provide a standard last sentence (lines 412 - 415) about why this study is important.  Further detail would enhance this.

Response 6: Upon general review of the discussion, current study was added to the reference list and the contents were revised accordingly. In addition, I have added the ‘6. Impact Statement’ section of the manuscript. Thank you again for such detailed feedback.

Sincerely.

Round 2

Reviewer 2 Report

Satisfied to accept the manuscript